# Microbial Diversity, Co-Occurrence Patterns, and Functional Genes of Bacteria in Aged Coking Contaminated Soils by Polycyclic Aromatic Hydrocarbons: Implications to Soil Health and Bioremediation

**DOI:** 10.3390/microorganisms13040869

**Published:** 2025-04-10

**Authors:** Liping Zheng, Yifan Yan, Qun Li, Junyang Du, Xiaosong Lu, Li Xu, Qunhui Xie, Yangsheng Chen, Aiguo Zhang, Bin Zhao

**Affiliations:** 1State Key Laboratory of Environmental Chemistry and Ecotoxicology, Research Center for Eco-Environmental Sciences, Chinese Academy of Sciences, Beijing 100085, China; lpzheng_st@rcees.ac.cn (L.Z.); lixu@rcees.ac.cn (L.X.); qhxie@rcees.ac.cn (Q.X.); 2University of Chinese Academy of Sciences, Beijing 100049, China; 3Nanjing Institute of Environmental Sciences, Ministry of Ecology and Environment, Nanjing 210042, China; yanyifan88@163.com (Y.Y.); liqun@nies.org (Q.L.); hjgcdjy@163.com (J.D.); luxiaosong2014@163.com (X.L.); 4School of Environment, Hangzhou Institute for Advanced Study, UCAS, Hangzhou 310024, China; binzhao@ucas.ac.cn

**Keywords:** PAHs, coking plant, microbial diversity, co-occurrence patterns, functional gene

## Abstract

PAH contamination from coking plants have received widespread attention. However, the microbial diversity, co-occurrence patterns, and functional genes of bacteria in aged coking contaminated soils by PAHs are still not clear. In our study, we used a macro-genetic approach to detect PAH-contaminated soils from both a coking production area (CA group) and an office zone (OA group) in an abandoned coking plant, and analyzed the characteristic bacteria and function genes, microbial network interaction patterns, and soil P-cycling in long-term PAH-contaminated soils. The results revealed that *Proteobacteria* were significantly positively correlated with PAHs and *Betaprobacteria bacterium* rifcsplowo2 12 full 6514, candidatus *Muproteobacteria bacterium* RBG16609, and *Sulfurifustis variabilis*, which belong to *Proteobacteria*, were characteristic bacteria in PAH-contaminated soils. The phn, which is the PAH degradation gene, was abundantly expressed in the PAH-contaminated soil. The *phn* gene cluster genes (*phnE*, *phnC*, and *phnD*) were significantly expressed in the CA group of PAH-contaminated soils (*p* < 0.05). By integrating microbial diversity, network structure, and functional genes, it offers a comprehensive understanding of soil ecosystem response indicators to prolonged PAH stress. The results of this study will provide new ideas for constructing an assessment index system for soil health and screening biomarkers for PAH-contaminated soils.

## 1. Introduction

Polycyclic aromatic hydrocarbons (PAHs) are a class of organic compounds composed of two or more condensed benzene rings. PAHs in the environment can enter the human body through various pathways, so the treatment and remediation of PAH pollution has received widespread attention. PAH contamination in the soil environment comes from a wide range of sources, among which the dispersal or seepage of petroleum compounds containing PAHs from coking plants is one of the most important sources [1,2,3].

PAHs are divided into two categories: low molecular weight (LMW) compounds composed of fewer than four rings and high molecular weight (HMW) compounds of four or more rings. The higher the molecular weight of the PAHs, the more stable and hydrophobic they are [4]. So, they are not easily degradable and can remain in soil, sediment, and atmospheric particles for a long time, causing persistent environmental pollution. Considering the economic benefits and high efficiency of microbial remediation, it has become an ideal method and an important means of removing PAHs from the environment [5]. Researchers have isolated a variety of PAH-degrading bacteria from the environment, including *Pseudomonas*, *Rhodococcus*, *Aeromonas*, *Bacillus*, *Burkholderia*, *Mycobacterium*, and *Sphingomonas* etc. [6], but there are not many efficient degrading strains covered in them, especially the high-ring PAH-degrading bacteria, which are even rarer. Therefore, it is of great practical significance to screen for efficient PAH-degrading bacteria or mixed strains.

In this study, we used gene sequencing to perform a macro-genomic analysis of bacterial communities in aged PAH-contaminated soil from an abandoned coking plant. We analyzed the correlations between the soil bacterial community and PAHs. The main objective of this study was to investigate the structure and characteristic genes of soil bacterial communities with long-term PAH contamination, with special focus on (i) bacterial diversity and characteristic genes of PAH-contaminated soils; (ii) soil microbial interactions in soils from the coking plant; (iii) the effect of PAH contamination on the function of soil microbial-mediated phosphorus cycling. The changes in the soil community structure and microbial functional genes under long-term stress of PAHs were investigated in the soil of an abandoned coking plant. The results of this study will provide new ideas for constructing an assessment index system for soil health and screening biomarkers for PAH-contaminated soils.

## 2. Materials and Methods

### 2.1. Sample Collection

Our study investigated an old, abandoned coking plant in eastern China (Figure 1). To meet the needs of urbanization and development, the coking plant ceased production in 2015, and most of the production equipment in the plant has been dismantled. Soil samples were collected following HJ/T166-2004 [7] We collected four soil samples at the coking production area (CA, S1–S4) and four soil samples at the office area (OA, S6–S8) of the site in the winter of 2022. Each sample was a mixture of five soil cores (0–15 cm) from a 1 × 1 m^2^ area at the sampling point. The soil samples were collected in sterile bottles insulated with dry ice and sent to the laboratory to be stored at −80 °C for microbiological and soil chemical analysis.

### 2.2. PAH Analysis

The 16 PAHs prioritized for control by the U.S. EPA were extracted according to the standard method (HJ834-2017) [8]. A total of 10 g of soil samples were weighed and diatomaceous earth was added to grind them into a fine powder. The resulting sample was then transferred to an ASE accelerated solvent extractor (E-916, Stepqi Laboratory Instruments Ltd., Uster, Switzerland), and 6 mix surrogates (2-fluorophenol, 2-fluorobiphenyl, phenol-D6, nitrobenzene-D5, 2,4,6-tribromophenol, and p-tertiary-D14) were added, and a mixture of dichloromethane and acetone (1:1, V:V) was used for the extraction. After extraction and filtration, the concentration was concentrated to 1 mL in a water bath, and then the internal standard (a mixture of six deuterated PAHs) was added. PAHs were analyzed by gas chromatography–mass spectrometry (GC–MS) (7890B-5977B, Agilent Technologies, Inc., Santa Clara, CA, USA) and qualitatively analyzed based on ion fragmentation and retention time, and quantified by the quantitative ion internal standard method. Quality control follows HJ834-2017. Laboratory blanks and laboratory parallels were used with 6 mix surrogates (2-fluorophenol, 2-fluorobiphenyl, phenol-D6, nitrobenzene-D5, 2,4,6-tribromophenol, and p-tertiary-D14), and the standards used for QC consisted of 6 mix surrogates and 64-component semivolatile mixture standards.

### 2.3. DNA Extraction and Metagenome Sequencing Analysis

The total DNA was extracted from the soil using a Magnetic Soil And Stool DNA Kit (TIANGEN, Beijing, China) according to the manufacturer’s instructions. The DNA sample was fragmented by sonication to a size of 350 bp, then DNA fragments were end-polished, A-tailed, and ligated with the full-length adaptor for Illumina sequencing with further PCR amplification. At last, PCR products were purified using the AMPure XP system, and libraries were analyzed for size distribution by the Agilent 2100 Bioanalyzer and quantified using real-time PCR (Bio-Rad CFX96, Hercules, CA, USA). The clustering of the index-coded samples was performed on a cBot Cluster Generation System according to the manufacturer’s instructions. After cluster generation, the library preparations were sequenced on an Illumina Novaseq 6000 (Illumina, San Diego, CA, USA), generating 150 bp paired-end reads. The data generated from the Illumina (or BGI) platform were used for bioinformatics analysis. The extraction and sequencing methods of total DNA are described in Appendix A. All of the analyses and quality control were performed by Shanghai Applied Protein Technology (Shanghai, China).

### 2.4. Data Analysis

SPSS Statistical Package 17.0 (IBM, New York, NY, USA) and Excel 2020 (Microsoft, Redmond, WA, USA) were used to process the data. The network was visualized in Gephi version 0.10.1 [9]. R software (3.5.3 version) was used to carry out PCoA and Lefse (*p* < 0.05 and an LDA score > 3) analyses. The R language package(PCoA: stats 4.2.2 and LEfSe 1.1.01) is in the Appendix A.

## 3. Results

### 3.1. Concentration Distributions of PAHs

The highest content of total PAHs was found in S3 (554.05 mg/kg), and the lowest was in S7 (3.27 mg/kg) (Appendix A). We used the Mann–Whitney test to analyze the PAHs in two areas (coking production area and office area), the result showed that there was a significant difference (*p* < 0.05) between the two areas (Appendix A). Additionally, the PAHs in all the samples were dominated by high molecular weight (HMW) PAHs, accounting for 65.72% to 83.86% of the total (Figure 2).

### 3.2. Relationship Between Bacterial Communities and PAHs

Both LMW-PAHs and HMW-PAHs were significantly positively correlated with *Proteobacteria* and *Aciddobacteria* (*p* < 0.05) (Figure 3). Furthermore, *Proteobacteria* showed a positive correlation with *Chloroflexi* and *Gemmatimonadetes* (*p* < 0.05). Principal coordinates analysis (PCoA) of the soil communities revealed that the similarities between the phylum types of CA and OA were significantly different (Appendix A). The findings reveal the existence of characteristic microorganisms that have adapted to this stressful environment in PAH-contaminated soil, and the results of the study indicate the characteristics of these microbial phyla in the biodegradation of soil PAHs.

There were significant differences between the CA and OA groups at the species level (*p* < 0.05) (Appendix A). Lefse analysis identified significantly different species markers in the CA group (Figure 4), primarily comprising of the following: *Chloroflexi bacterium* RGB 16070013/167214, *Gemmatimonadetes bacterium*, *Betaprobacteria bacterium* rifcsplowo2 12 full 6514, candidatus *Muproteobacteria bacterium* RBG16609, and *Sulfurifustis variabilis*. These findings are consistent with the results of our study.

### 3.3. Co-Occurrence Network Analysis

The network analysis was used to explore the co-occurrence patterns in soil microbial communities (Figure 5). The co-occurrence network had a total of 199 nodes and 2356 edges, with a modularity index of 3.74 (a value > 0.4 indicates that the network has a modular structure). The highly related microorganisms in the soil bacterial co-occurrence network were mainly categorized into three larger modules (Figure 5a), accounting for 41.71%, 36.68%, and 15.58%. The co-occurrence network nodes were mainly *Proteobacteria* (40.1%), *Acidobacteria* (7.11%), *Actinobacteria* (31.98%), *Chloroflexi* (5.58%), and *Gemmatimonadetes* (4.06%) (Figure 5b). According to the betweenness centrality of the association network [10], *Phycicoccus_jejuensis*, *Nocardioides*_sp_LS1, *Cyanobacteria_bacterium*_13_1_40 CM_2_61_ 4, and *Nitriliruptor_alkaliphilus* were identified as key taxa, suggesting that these microorganisms play a key role in the co-occurrence network.

### 3.4. Functional Gene of Soil Phosphorus Cycling and PAH Degradation

The soil phosphorus cycle is a key component of the Earth’s biogeochemical cycles and is essential for maintaining ecosystem function, supporting plant growth, and ensuring the healthy functioning of the global food chain [11,12]. Soil microorganisms are able to make phosphorus more readily available for plant uptake by solubilizing insoluble phosphorus compounds [13]. At the same time, microorganisms are also able to increase the bioavailability of phosphorus by converting organic phosphorus into an inorganic form, a process known as mineralization.

Principal coordinates analysis (PCoA) was performed on the microbiological phosphorus functional genes of the studied soils (Appendix A). There was a significant difference in the first principal component between the CA and OA groups, indicating that the phosphorus functional genes of soil microorganisms in the two groups differed. The differential genes of the two groups based on relative abundance (Mann–Whitney–Wilcoxon test, Figure 6) showed that the *phoD* functional gene in the OA group was significantly higher (*p* < 0.05). The *phoD* gene is widely reported among the microorganisms involved in the soil phosphorus cycle [11], and these microorganisms play an important role in soil phosphorus conversion and utilization by secreting alkaline phosphatase. Phosphatase is concerned with hydrolyzing organic phosphorus compounds in the soil phosphorus cycle and converting them into inorganic forms of phosphorus that can be absorbed and utilized by plants, thus increasing the bioavailability of soil phosphorus [14]. The results of our study showed that soil alkaline phosphatase gene abundance was higher in the OA group, which could be because the phosphorus metabolic cycling ability of soils in the OA group was better. The metabolic cycling of soil phosphorus was inhibited in the CA group.

Meanwhile, the relative abundance of *phnP*, *phnE*, *phnC*, and *phnD* genes, which belong to the *phn* gene cluster, in CA was significantly higher (*p* < 0.05). The *phn* gene cluster comprises a group of tightly spaced genes that are relatively centralized on the chromosome, and collectively encode the enzymes and auxiliary proteins involved in the degradation of aromatic compounds [15]. It has been reported that the *phn* genes of the burkholderia cepacia RP007 strain constitute a cluster of genes involved in the catabolism and metabolism of polycyclic aromatic hydrocarbons. These genes are transcribed and the recombinant *phn* enzyme is able to oxidatively metabolize naphthalene and phenanthrene when the RP007 strain needs to consume them in its growth process [16]. Researchers have identified the genes for the entire catabolic pathway of phenanthrene through the phylogeny of the cyclic hydroxylated dioxygenase (RHD) that initiates bacterial PAH metabolism on a new genomic island (GEI) approximately 232 kb long, referred to as the *phn* island [17]. The presence or absence of the *phn* gene and the level of its expression can also be used as an indicator of an environmental pollutant’s degradation capacity. By detecting the expression of the *phn* gene in a specific environment, the degradation capacity of the environment for PAHs and chlorinated aromatic compounds can be assessed, providing a scientific basis for environmental monitoring and assessment.

## 4. Discussion

Macro-genome sequencing was performed on soils from an abandoned coking plant to understand the soil community structure and biomarkers under long-term PAH stress. The results of the microbial community composition were similar to those of Shuying Geng [7] in the soil of an abandoned chemical plant. Geng found that the dominant phyla in all samples were responsible for PAH degradation and included *Proteobacteria* (20.86–81.37%) and *Chloroflexi* (2.03–28.44%) [7]. Tang reported that microbial diversity analysis of PAH-contaminated deep soil from a chemical plant showed that the dominant phyla degrading PAHs were *Pseudomonas* and *Acinetobacter* [18]. The genera *Enterobacteria* and *Pseudomonas*, which belong to the gamma-proteobacteria, played a major role in the degradation of PAHs in a coal tar-contaminated soil [19]. Our study found that PAHs were significantly positively correlated with *Proteobacteria* and *Acidobacteria* (*p* < 0.05). The results indicated the presence of the characteristic bacteria in long-term PAH-contaminated soils to adapt to the long-term stress of PAHs and for soil self-purification. Meanwhile, soil microflora is a complex network interaction system, and the results showed that the *Phycicoccus_jejuensis*, *Nocardioides*_sp_LS1, *Cyanobacteria_bacterium*_13_1_40 CM_2_61_4, and *Nitriliruptor_ alkaliphilus* were key taxa. Similarly, Yiyi Zhao found that key species may influence microbial assemblage patterns through synergistic interactions [20]. Research has shown that PAH-degrading bacteria (PDB) from various depths of a chemical waste land soil belonged to Actinomycetes and Proteobacteria, which dominated the bacterial communities during the early and full stages of the biopurification process [21]. Keystone species can influence the composition and aggregation of the microbial communities [22,23]. Many studies have reported that the keystone species are likely to play important roles in caring for microbial network structure of soil [24,25,26]. Briefly, the biodegradation of PAHs requires interactions and co-adaptation between soil bacterial communities [27,28].

Microorganisms generally produce a series of metabolic reactions through the expression of specific degradation genes and degradation enzymes, which in turn lead to the transformation and degradation of PAHs [29]. Degradation genes encode enzymes, mainly including dioxygenases, hydratases, aldolases, and dehydrogenases [30]. According to the current study, microorganisms have PAH degradation genes, mainly including *nid*, *phd*, *nah*, *phn*, and *pdo*. In the present study, it was found that the *phn* gene was abundantly expressed in the PAH-contaminated soil of the coking area, and the *phn* gene mainly encodes a dioxygenase, which achieves the carboxylation of the aromatic ring for the degradation of naphthalene, phenanthrene, pyrene, and other PAHs [31,32,33,34].

What is a healthy soil? Professor Larkin [35] proposed the concept of soil health, which includes an indicator as soil having some resistance and resilience to environmental degradation and stress. The Cornell Soil Health Evaluation System [36] also suggests that healthy soils should be easy to farm and can be resistant to unfavorable environments. Maintaining soil health, exploring soil multi functionality, and developing soil green products are the keys to enhancing sustainable soil development and human One Health unification [37]. In the present study, soil microorganisms adopted adaptive strategies to survive under prolonged and high concentrations of PAHs, and evolved characteristic bacterial communities [38,39,40,41]. Healthy soil should be uncontaminated and rich in beneficial microbial species involved in nutrient cycling and organic matter decomposition functions. In contrast, the microbial communities of the polluted soil in the coking plant were changed, which provides a reference for the index system of soil health evaluation. Furthermore, the evaluation system of soil health should include biological indicators such as the soil microbial community structure of healthy soil.

## 5. Conclusions

This study provides new insights into the structural characteristics of soil bacterial communities under long-term stress from PAHs. *Proteobacteria* were significantly positively correlated with PAHs and *Betaprobacteria bacterium* rifcsplowo2 12 full 6514, candidatus Muproteobacteria bacterium RBG16609, and Sulfurifustis variabilis, which belong to *Proteobacteria*, were characteristic bacteria in PAH-contaminated soils. The *phn*, which is the PAH degradation gene, was abundantly expressed in the PAH-contaminated soil. By integrating microbial diversity, network structure, and functional genes, it offers a comprehensive understanding of soil ecosystem response indicators to prolonged PAH stress. The findings of this study will provide new perspectives for constructing an assessment index system for soil health and screening biomarkers for PAH-contaminated soils at coking plants.

## Figures and Tables

**Figure 1 microorganisms-13-00869-f001:**
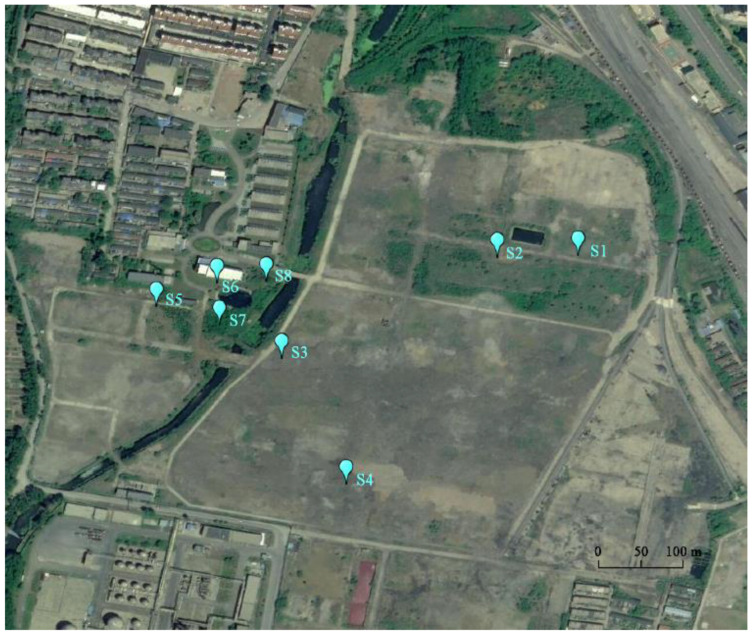
Remote sensing image of the study area.

**Figure 2 microorganisms-13-00869-f002:**
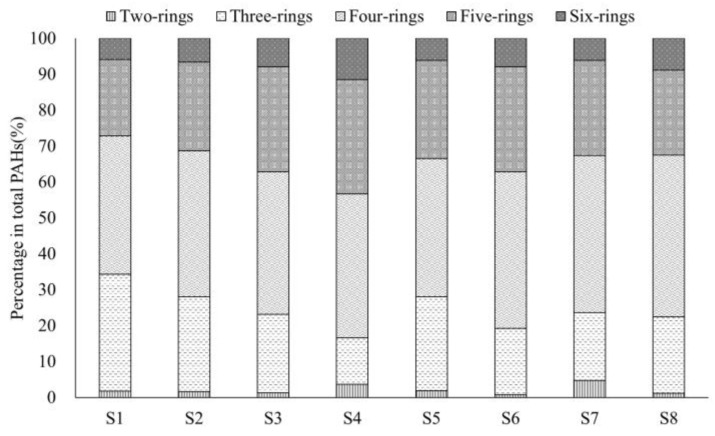
Distribution of PAHs depending on the number of rings in the molecule.

**Figure 3 microorganisms-13-00869-f003:**
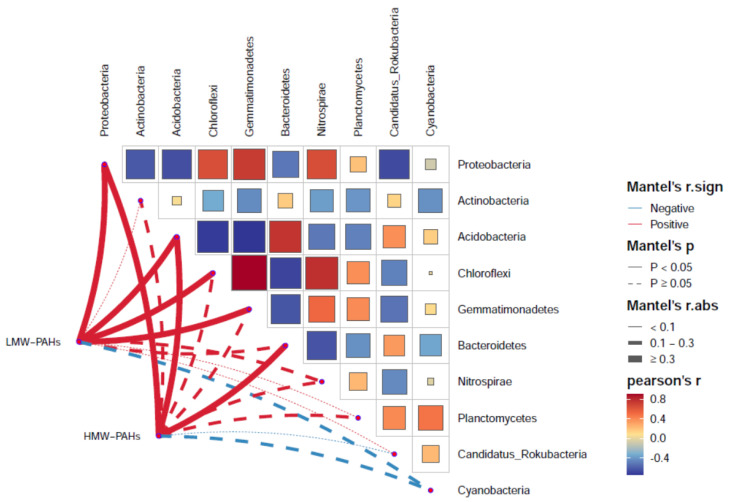
Correlation between the top 10 phylum and PAHs.

**Figure 4 microorganisms-13-00869-f004:**
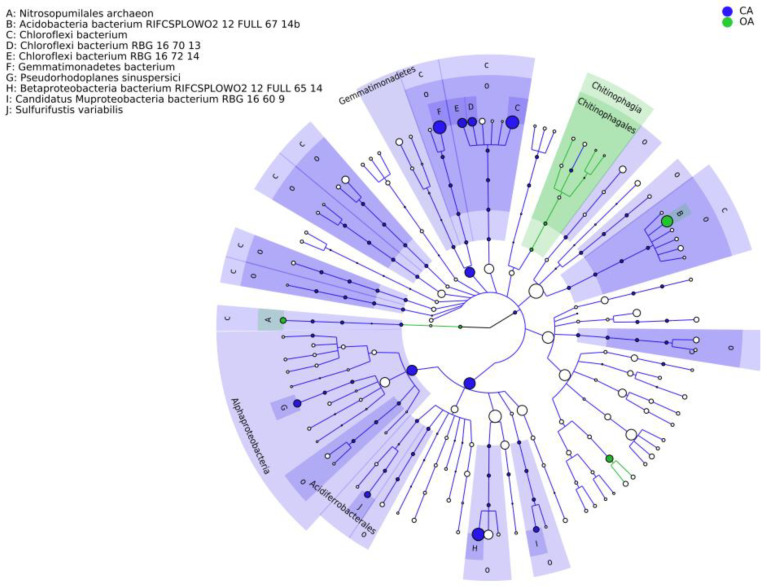
Annotated branching map of different species.

**Figure 5 microorganisms-13-00869-f005:**
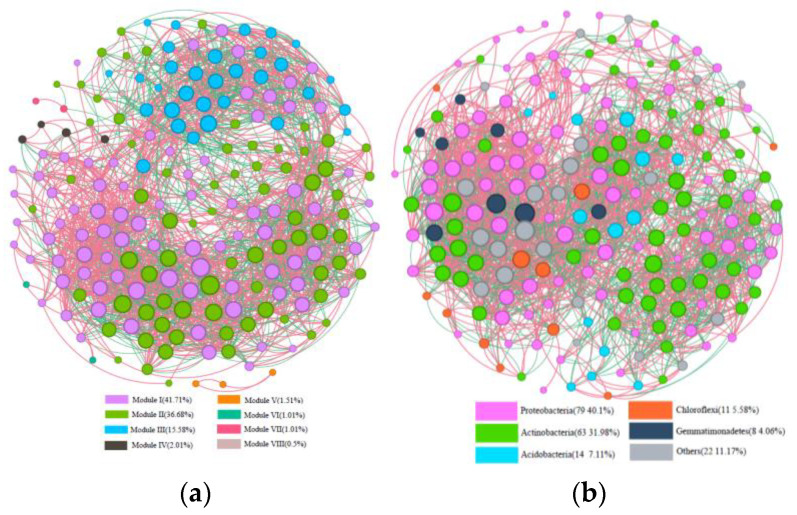
Co-occurrence networks of microbial communities. (**a**) Different modularity classes; (**b**) node colors indicate different phyla. The red line means positive correlation and the green line means negative correlation.

**Figure 6 microorganisms-13-00869-f006:**
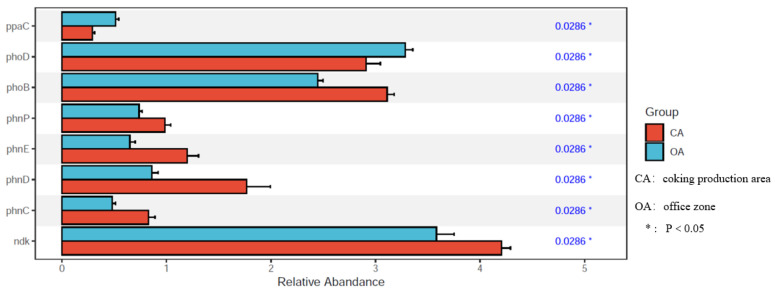
Significantly different functional genes.

## Data Availability

The original contributions presented in this study are included in the article. Further inquiries can be directed to the corresponding author.

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
