# Peer review of "Microbial Diversity, Co-Occurrence Patterns, and Functional Genes of Bacteria in Aged Coking Contaminated Soils by Polycyclic Aromatic Hydrocarbons: Implications to Soil Health and Bioremediation"

_microorganisms, 2025, doi:10.3390/microorganisms13040869_

Round 1

Reviewer 1 Report

Comments and Suggestions for Authors

The authors of the study “Microbial Diversity, Co-Occurrence Patterns and Functional Gene of Bacterial in Aged Coking Contaminated Soils by PAHs: Implications to Soil Health and Bioremediation” make an interesting approach to microbial diversity and co-occurrence patterns in aged coking contaminated soils by PHAs. However, I have to make a number of comments:
1. In the abstract, define the CA group (it has not been defined before and, therefore, the reader cannot know what the acronym corresponds to) or not refer to the site as an acronym.
2. The authors should rewrite the sentence on lines 42-44 to express the idea more clearly. Also, include something like “with a number of XXX” when defining LMW and HMW.
3. Why do the authors focus on the phosphorus cycle and not (or why not also) on any other biogeochemical cycle?
4. In section 2.3. the authors should include what type of libraries they generate, how many reads they obtain, what criteria they follow for trimming, what type and criteria of assembly and indicate in which public repository they have deposited the sequences generated in the study.
5. Do the authors carry out the PCoA and Lefse analyses (which should also be defined) in SPSS and/or Excel? If not, indicate which computer tools they have used as well as the parameters with which these analyses have been carried out.
6. The sentences in lines 112-115, 126-128 and 143-148 correspond to discussion, so please move them there.
7. In lines 157-159 the authors indicate that microorganisms increase the bioavailability of phosphorus by converting it into inorganic forms (mineralization). Is this really the case?
8. Please name the gene names in italics (phoD on line 165, phnP, phnE (etc) on line 175, among others)
9. The authors do not perform any functional validation, so they cannot assure that the higher abundance in the alkaline phosphatase gene indicates a higher activity of the phosphorus cycle (line 173). Please replace “indicating” with “could be” or something similar.
10. Define, in figure 6, CA and OA so that it is self-contained.
11. Please always follow the same format when citing references (either with or without spaces) (line 198 vs line 200). Check the rest.
12. It would be necessary, to support the sequencing results, the possible differences in the bioremediation capacity of the different sampling sites.
13. In the references section, use the same format. For example, whether or not to use a comma after naming the journal (ref. 25 vs. ref. 26). Check the rest.

Author Response

Comments 1: In the abstract, define the CA group (it has not been defined before and, therefore, the reader cannot know what the acronym corresponds to) or not refer to the site as an acronym.

Responds 1: The CA group has been defined in the abstract as follows:

In our study, we used a macro-genetic approach to detect PAHs-contaminated soils from both production areas (CA group) and office zones (OA group) in an abandoned coking plant.

Comments 2: The authors should rewrite the sentence on lines 42-44 to express the idea more clearly. Also, include something like “with a number of XXX” when defining LMW and HMW.

Responds 2: The sentence are rewrite as follows:

PAHs are divided into two categories: low molecular weight(LMW) compounds composed of fewer than four rings and high molecular weight(HMW)compounds of four or more rings. The higher the molecular weight of PAHs, the more stable and hydrophobic they are, so they are not easily degradable and can remain in soil, sediment and atmospheric particles for a long time, causing persistent environmental pollution.

Comments 3: Why do the authors focus on the phosphorus cycle and not (or why not also) on any other biogeochemical cycle?

Responds 3: Soil phosphorus is one of the essential nutrients for plant growth and microbial activities, and can be coupled with carbon and nitrogen to participate in biogeochemical cycles. Soil phosphorus is crucial for maintaining ecosystem services, structure and function, and is one of the research hotspots in recent years. Based on the consideration of research hotspot and importance, we focus on the phosphorus cycle.

Comments 4: In section 2.3. the authors should include what type of libraries they generate, how many reads they obtain, what criteria they follow for trimming, what type and criteria of assembly and indicate in which public repository they have deposited the sequences generated in the study.

Responds 4:DNA libraries were generated. The reads can be found in the table in supplementary data. The reads of all the samples were merged and assembled using MEGAHIT, a splicing software based on the principle of De-Brujin graph, to construct the De-Brujin graph based on the overlap relationship between kmer, to obtain the contigs, and to filter the contigs with more than 800 bp for counting and for subsequent analysis. Contigs above 800 bp were screened and used for subsequent analysis. The following has been added in section 2.3 :The extraction and sequencing methods of total DNA are described in supplementary data. (Table is in the attachment file)

Comments 5: Do the authors carry out the PCoA and Lefse analyses (which should also be defined) in SPSS and/or Excel? If not, indicate which computer tools they have used as well as the parameters with which these analyses have been carried out.

Responds 5:The following has been added in section 2.4 :R software (3.5.3 version) were carried out to PCoA and Lefse (p < 0.05 and an LDA score >3) analyses. The R language package is in supplementary data.

Comments 6: The sentences in lines 112-115, 126-128 and 143-148 correspond to discussion, so please move them there.

Responds 6:The sentences in lines 112-115, 126-128 and 143-148 have been moved to discussion.

Comments 7: In lines 157-159 the authors indicate that microorganisms increase the bioavailability of phosphorus by converting it into inorganic forms (mineralization). Is this really the case?

Responds 7: Soil microorganisms increase the bioavailability of phosphorus in the soil by decomposing organic phosphorus compounds in the soil and converting them to inorganic phosphorus, which helps plants to absorb soil phosphorus components. With references: (1) ZHANG Wannian, YANG Zi, YAN Yupeng, WANG Xiaoming, et al. Research Progress on Soil Organic Phosphorus Mineralization and Its Regulation[J]. Acta Pedologica Sinica,2025,62(2):334-347.(2) Turner B L, Frossard E, Baldwin D S. Organic phosphorus in the environment[M]. Wallingford, Oxfordshire:CABI Publishing, 2005.
    Comments 8: Please name the gene names in italics (phoD on line 165, phnP, phnE (etc) on line 175, among others)

Responds 8:The phoD on line 165, phnP, phnE (etc) on line 175, among others have been named in italics.

Comments 9: The authors do not perform any functional validation, so they cannot assure that the higher abundance in the alkaline phosphatase gene indicates a higher activity of the phosphorus cycle (line 173). Please replace “indicating” with “could be” or something similar.

Responds 9:The “indicating” in line 173 has been replaced with “could be”.
Comments 10: Define, in figure 6, CA and OA so that it is self-contained.

Responds 10: CA and OA have been defined in figure 6.
Comments 11: Please always follow the same format when citing references (either with or without spaces) (line 198 vs line 200). Check the rest.

Responds 11: The references in the manuscript have been followed the same format as line 198.
Comments 12:  It would be necessary, to support the sequencing results, the possible differences in the bioremediation capacity of the different sampling sites.

Responds 12:Since the Mann-Whitney-Wilcoxon test requires a sample size (n) of n ≥ 3 for each group, and there were four sampling points for each of the CA (cooking production area) and OA (office zone) groups, the difference in bioremediation capacity between the two groups was statistically derived. This is the preliminary result of our study, if we want to know more about the bioremediation capacity of the different sampling sites in CA (cooking production area), I suggest that we can design the next sampling to collect more samples specifically for the CA group, to investigate the bioremediation capacity of the soil microorganisms in this area,so that the bioremediation capacity of soil microorganisms can be studied specifically.

Comments 13: In the references section, use the same format. For example, whether or not to use a comma after naming the journal (ref. 25 vs. ref. 26). Check the rest.

Responds 13: References have all been revised to a format without commas.

Reviewer 2 Report

Comments and Suggestions for Authors

In this Article, the authors considered the study of the macro-genetic approach to detect PAHs-contaminated soils in coking plants, analyzing the characteristic bacteria and function genes, microbial network interaction patterns, and soil P-cycling in PAHs long-term contaminated soils.

The Article in general is well written and organized, with a lot of processed data. Before publication, manuscript needs minor revision.

  1. In the final paragraph of the Introduction part, the novelty of this study must be highlighted.
  2. Suggest adding description of the current status of the local society, urbanism, traffic, industry, agriculture, in the study area, and relevance of this study for the environment.
  3. Sampling year and time interval (winter or summer), should be added.
  4. Which guidelines/standards were followed for the soil samples collection and PAHs analysis? It would be necessary to add that information’s.
  5. Please add quality control and analysis part.
  6. It would be desirable to add GC-MS chromatograms to the manuscript, in order to gain a better insight into the PAHs-pollution level of the study area.
  7. In lines 137 and 140 replace Fig. 4a and Fig. 4b with Fig. 5a and Fig. 5b.
  8. Please compare results of the present study with other previously published literature.
  9. What is the future perspective of this study? Will bioremediation of the existing pollution be carried out and in what way?
  10. Recommendation is to ignore some old citations and add more recent researches, in order to represent the importance and latest findings in this research field.

Author Response

Comments 1: In the final paragraph of the Introduction part, the novelty of this study must be highlighted.

Responds 1: Highlight had been added in the final paragraph of the Introduction part: The changes in soil community structure and microbial functional genes under long-term stress of PAHs were investigated in the soil of an abandoned coking plant. The results of this study will provide new ideas for constructing an assessment index system for soil health and screening biomarkers for PAH-contaminated soils.

Comments 2: Suggest adding description of the current status of the local society, urbanism, traffic, industry, agriculture, in the study area, and relevance of this study for the environment.

Responds 2: We have added a background description of the study area in section 2.1:For the need of urbanization and development, the coking plant ceased production in 2015, and most of the production equipment in the plant has been dismantled.

Comments 3: Sampling year and time interval (winter or summer), should be added.

Responds 3: We have added sampling year and time:We collected four soil samples at the coking production area (CA, S1-S4) and four soil samples at the office area (OA, S6-S8) of the site in the winter of 2022.

Comments 4: Which guidelines/standards were followed for the soil samples collection and PAHs analysis? It would be necessary to add that information’s.

Responds 4: HJ/T166-2004 was followed for the soil samples collection,HJ834-2017 was followed for the PAHs analysis. The guidelines/standards have been added in section 2.1 and 2.2.

Comments 5: Please add quality control and analysis part.

Responds 5:We have added quality control and analysis part in section2.2.

Comments 6: It would be desirable to add GC-MS chromatograms to the manuscript, in order to gain a better insight into the PAHs-pollution level of the study area.

Responds 6: Since there were a total of 8 sites 3 parallels for each site, the GC-MS chromatograms of a total of 24 samples. We have selected some chromatograms and put them in the appendix

Comments 7: In lines 137 and 140 replace Fig. 4a and Fig. 4b with Fig. 5a and Fig. 5b.

Responds 7: Fig. 4a and Fig. 4b have been replaced with Fig. 5a and Fig. 5b.

Comments 8: Please compare results of the present study with other previously published literature.

Responds 8:In the discussion section, previously published literature is presented and compared with the study in this paper.

Comments 9: What is the future perspective of this study? Will bioremediation of the existing pollution be carried out and in what way?

Responds 9:Using actual soil as a study object to screen biomarkers for PAH-contaminated soils and further development of bioremediation technologies for PAHs-contaminated soil is the future prospect of this research. I believe that we are currently at the stage of research on microbiological indicators of PAH-contaminated soils, and the existing remediation of contamination needs to be finalized only after comprehensive consideration of physical, chemical, and biological measures, as well as time and economic costs.

Comments 10:Recommendation is to ignore some old citations and add more recent researches, in order to represent the importance and latest findings in this research field.

Responds 10: New references have been added:Reference 36-39.

Reviewer 3 Report

Comments and Suggestions for Authors

The authors of the paper entitled “Microbial Diversity, Co-Occurrence Patterns and Functional Gene of Bacterial in Aged Coking Contaminated Soils by PAHs: Implications to Soil Health and Bioremediation” have done an interesting work. In their work they have critically analysed the characteristic bacteria and function genes, microbial network interaction patterns, and soil P-cycling in PAHs long-term contaminated soils, they used macro-genetic approach to detect PAHs-contaminated soils in coking plants. . They inferred that by integrating microbial diversity, network structure, and functional genes, it offers a comprehensive understanding of soil ecosystem response indicators to prolonged PAHs stress. The results of this study will provide new ideas for constructing an assessment index system for soil health and screening biomarkers for PAH-contaminated soils. All in all the word has been well researched and detail investigated. I would suggest acceptance of the paper after minor revision. The authors need to compare their work with similar studies and need to have a deeper discussion on the novelty of their work Additional there are lots of English grammatical and fluency problems that need to be edited and revised with fluent anguish writers.

Author Response

Comments 1: The authors of the paper entitled “Microbial Diversity, Co-Occurrence Patterns and Functional Gene of Bacterial in Aged Coking Contaminated Soils by PAHs: Implications to Soil Health and Bioremediation” have done an interesting work. In their work they have critically analysed the characteristic bacteria and function genes, microbial network interaction patterns, and soil P-cycling in PAHs long-term contaminated soils, they used macro-genetic approach to detect PAHs-contaminated soils in coking plants.  They inferred that by integrating microbial diversity, network structure, and functional genes, it offers a comprehensive understanding of soil ecosystem response indicators to prolonged PAHs stress. The results of this study will provide new ideas for constructing an assessment index system for soil health and screening biomarkers for PAH-contaminated soils. All in all the word has been well researched and detail investigated. I would suggest acceptance of the paper after minor revision. The authors need to compare their work with similar studies and need to have a deeper discussion on the novelty of their work Additional there are lots of English grammatical and fluency problems that need to be edited and revised with fluent anguish writers.

Responds 1: A comparison with similar studies the novelty of the work has been added in section 4. We have added some new references and discussed our work for soil health and biomarkers.
